# Microbiota, IgA and Multiple Sclerosis

**DOI:** 10.3390/microorganisms10030617

**Published:** 2022-03-14

**Authors:** Léo Boussamet, Muhammad Shahid Riaz Rajoka, Laureline Berthelot

**Affiliations:** 1Centre for Research in Transplantation and Translation Immunology, Nantes Université, Inserm, CR2TI UMR, 1064 Nantes, France; leo.boussamet@etu.univ-nantes.fr; 2Laboratory of Animal Food Function, Graduate School of Agricultural Science, Tohoku University, Sendai 980-8572, Japan; shahidrajoka@yahoo.com

**Keywords:** IgA, gut microbiota, multiple sclerosis, neuroinflammation, regulation, IL-10

## Abstract

Multiple sclerosis (MS) is a neuroinflammatory disease characterized by immune cell infiltration in the central nervous system and destruction of myelin sheaths. Alterations of gut bacteria abundances are present in MS patients. In mouse models of neuroinflammation, depletion of microbiota results in amelioration of symptoms, and gavage with MS patient microbiota exacerbates the disease and inflammation via Th17 cells. On the other hand, depletion of B cells using anti-CD20 is an efficient therapy in MS, and growing evidence shows an important deleterious role of B cells in MS pathology. However, the failure of TACI-Ig treatment in MS highlighted the potential regulatory role of plasma cells. The mechanism was recently demonstrated involving IgA+ plasma cells, specific for gut microbiota and producing IL-10. IgA-coated bacteria in MS patient gut exhibit also modifications. We will focus our review on IgA interactions with gut microbiota and IgA+ B cells in MS. These recent data emphasize new pathways of neuroinflammation regulation in MS.

## 1. Introduction

Multiple sclerosis (MS) is a complex inflammatory disease of the central nervous system (CNS) that causes a wide range of clinical symptoms, including physical and cognitive deficits. Despite its origin being unknown, the pathology involves both genetic and environmental factors [1], leading to immune cell infiltration and destruction of myelin sheaths and axons. This neurodegenerative disorder is characterized by sporadic abnormalities and gradual neurodegeneration and is triggered by complex, dynamic interplay between the immune system, glial cells, and neurons. New treatments targeting immune cells are now efficient in MS patients reducing relapse, lesion load in CNS, and delaying progression of the disease [2,3,4]. However, some patients with aggressive disease forms do not respond to the treatments, and patients with progressive disease forms still undergo accumulation of CNS lesions and disabilities. Therefore, the need for new therapies in MS is crucial and depends on a better understanding of pathological mechanisms occurring in MS. Since 2015, the first descriptions of gut microbiota dysbiosis in MS patients [5,6] have opened new perspectives for therapies in MS. In parallel, exploration of immune responses linked to microbiota modifications highlighted the potential deleterious and regulatory effects of different cell types. Investigations using germ-free (GF) mice have demonstrated the impact of the gut microbiota on MS experimental models. In experimental autoimmune encephalomyelitis (EAE), an experimental model of MS, the gut microbiota influences the immune response by affecting Th1-Th17/Th2 cell balance, Treg cells, and humoral immunity [7].

IgA B cell responses, at the close interface with microbiota, are important in this context and were recently shown to regulate neuroinflammation. In this review, we will develop the recent findings on microbiota alterations, IgA-coated microbiota, and IgA B cells in MS. The last part will be devoted to potential therapies targeting IgA/microbiota.

## 2. Gut Microbiota Alterations in Multiple Sclerosis

### 2.1. Gut-Brain Axis

With the discovery of numerous nerve endings in the intestine, the enteric nervous system has been considered the second brain [8]. Indeed, there is a close link between the brain and the gut consisting of a bidirectional communication system described as the gut-brain axis (summarized in Figure 1). Indeed, the central nervous system (CNS) has the ability to regulate intestinal motility as well as to orchestrate local immunity [9] via neuromediators involving the vagus nerve and the hypothalamic-pituitary-adrenal axis. In return, the digestive system can regulate components of the nervous system such as appetite [10] or mood [11]. These communications take place mainly through the neuroendocrine pathways, involving cytokines, neurotransmitters, and neuropeptides [12]. The immune system, interacting with the intestinal microbiota, is also a major player in these communications.

### 2.2. The Human Gut Microbiota: A Key Role in Maintaining Host Homeostasis

The human gut microbiota is constituted by trillions of microorganisms (bacteria, viruses, fungi, and other protozoa) living at the surface of the mucosa [13,14]. These microorganisms harbor 150 times more genes than the human genome and are essential for health [15]. Nonexistent at the fetal stage, the microbiota rapidly diversifies during infancy with bacteria that metabolize lactose in the first place. When solid food is introduced, a shift to carbohydrate, protein, and fat-metabolizing bacteria occurs [16].

In addition to its role as a barrier against pathogens, the intestinal microbiota participates in the production of essential nutrients such as vitamin K and vitamin B [17]. Indeed, germ-free rats and thus deprived of intestinal microbiota were shown to need greater intakes of these vitamins compared to mice raised in a normal environment [18,19]. Moreover, bacteria belonging to the *Firmicutes* phylum are able to produce short-chain fatty acids (SCFA) acetate (C_2_), propionate (C_3_), and butyrate (C_4_) as the main products of anaerobic fermentation, which represent the main source of energy for the colonic epithelial cells [20]. Bacteria living in the mucus layer also play a role in its maturation and recycling [21].

Another major role of the microbiota is in the modulation of the immune system and enteric nervous system, which are constantly stimulated and shaped by the microbial antigens [22,23]. Indeed, germ-free mice fed with sterile food exhibit altered enteric nervous systems compared to normal mice as well as altered immune responses (systemic T and B response deficiencies) [24,25], suggesting that exposure to microbial antigens is essential to educate a healthy immune system modulating both the innate and the adaptive immunity. Among these microbial compounds: SCFA secreted by some anaerobic bacteria were shown to harbor important modulatory properties toward the immune system. They appear to be major modulators of cytokine production (TNF-α, IL-2, IL-6, and IL-10) and migratory properties of leukocytes [26]. Pattern recognition receptors (PRRs), such as Toll-like receptors (TLRs), are important sensors of the microbiota present at the surface of epithelial cells and innate immune cells. For instance, Lipopolysaccharide (LPS), a key component of Gram-negative bacteria cell walls, creates a strong inflammatory response by macrophage and monocyte with production of IL-1β, TNFα, IL-6, and monocyte chemoattractant protein 1 (MCP-1) [27]. On the other hand, polysaccharide (PSA) arising from *Bacteroides fragilis* colonization activates anti-inflammatory genes in TLR1-TLR2-dependent way and drives naive CD4 T cell and B cells toward regulatory phenotypes (IL-10 and IL-12-producing cells) [28,29], attenuating inflammation. Moreover, certain strains of commensal Clostridia are known to be strong regulatory T cell inducers [30].

On the other hand, colonization by proinflammatory segmented filamentous bacteria promotes Th17 T cell differentiation and elicits the production of proinflammatory cytokines IL-17, IL-21, and IL-22 [31]. Finally, other compounds arising from bacterial activity, such as aryl hydrocarbon receptor (AhR) ligands or specific sphingolipids, are known to have regulatory effects on the immune system [32,33].

Under normal circumstances, these interconnections are finely regulated, and a balance between inflammation and regulation, response, and tolerance is maintained. Many environmental factors have been described as being able to modulate the microbiota composition. Among them, age, diet, or the use of certain medications are the main ones [34,35]. Long-term alterations in the microbiota/mucosal interface can result in systemic translocation of commensal microorganisms, susceptibility to pathogenic invasion, and chronic inflammatory immune responses. Disturbances of the microbiota leading to a pathological state constitute the dysbiotic state. Intestinal dysbiosis has been described in many inflammatory pathologies targeting a wide range of systems ranging from the gut with inflammatory bowel disease (IBD) [36,37] but was also observed in systemic diseases such as type 2 diabetes [38], lupus [39], or rheumatoid arthritis [40]. Recent studies highlighted that diseases affecting the CNS such as Parkinson’s and Alzheimer’s diseases [41,42], autism [43], or multiple sclerosis are also linked to gut dysbiosis to some extent. Indeed, the CNS is connected to the gut via sympathetic and parasympathetic nerves with close proximity to the microbiota, making it a potential target of interest both in exploring CNS disease mechanisms and as potential therapeutic leverage. The gut could be a relevant place to apply interventional therapeutics as molecules arising in the gut can have action on the CNS, either by retrograde axonal transport or by the circulatory system.

### 2.3. Alterations in Gut Microbiota of Multiple Sclerosis Patients

For the purpose of this review, we carried out a systematic review of all the reported studies investigating the gut microbiome content in multiple sclerosis. To identify studies of interest, we used the medical subject headings (MeSH) function in pubmed (https://pubmed.ncbi.nlm.nih.gov/, last accessed date: 31 December 2021). We interrogated the database using different term combinations with multiple sclerosis, microbiota, 16s rRNA, whole-genome sequencing and filtered for original articles. For the ongoing clinical trials, we searched through clinicaltrials.gov (https://clinicaltrials.gov/ accessed on 31 December 2021) database with multiple sclerosis in the condition input and terms such as probiotic, prebiotic, fecal microbiota transplant, short-chain fatty acids.

In multiple sclerosis (MS), we listed 36 studies investigating the gut microbiome content of MS patients ranging from 2015 to 2021, whose results are summarized in Table 1.

Although the vast majority of the studies involved case relapsing-remitting individuals versus healthy volunteers (HV), some of them also considered differences between subtypes of MS such as primary progressive MS (PPMS), secondary progressive MS (SPMS), and other related neurological diseases (neuromyelitis optica (NMO), clinically isolated syndrome (CIS)). Almost all studies, except for five (two Chinese studies, two Japanese and one multiethnic) looked at European ancestry individuals. Overall, four studies evaluated the effect of various treatments used in MS, and three studies evaluated microbiome modulation via dietary leverage in the context of MS. All studies except one evaluated the microbiome content from stool samples. While most of the research works investigated the microbiome content through 16S rRNA sequencing, four of them performed whole-genome sequencing. Only one of them investigated the fungal composition of the gut.

In terms of diversity, two components are widely used in the field of ecology. First, the α diversity index, mainly corresponding to the specific richness and Shannon indices, representing the number of species identified in a sample and the richness mitigated by the evenness, respectively. In the listed studies, α diversity showed contrasted results with two studies showing increased α diversity indices and three suggesting a decrease in the α diversity indices. The rest of the studies showed no significant results (NS) in this regard. Consequently, it is likely that there is no difference in the alpha diversity indices between MS and HV in their gut microbiota. Another considered index is β diversity. This index accounts for global differences between two groups of samples. In this case, 11 studies were able to show differences in β diversity index between MS and HV, suggesting an altered gut microbiome in MS with a global dysbiosis state. When looking at the taxonomic level, modification in the relative abundances of several bacteria occurs. Indeed, several genera were identified as impacted in several independent studies. Although results are highly variable, consensus seems to be emerging in some genera.

Among them, the decreased abundance in Firmicutes such as *Faecalibacterium prausnitzii* has been identified in 10 of the considered studies (one of them reported an increase). Moreover, decreased *Prevotella and Roseburia* genera were also reported (in 10 and 5 studies, respectively). On the other hand, increases in *Akkermansia*, *Streptococcus*, and *Blautia* were reported (in seven, six, and five studies, respectively).

While it is widely known that commensal bacteria can promote both inflammatory responses (Th1 and Th17) and regulatory (Th2) immune pathways, it seems that the balance between those two antagonist systems is broken in MS. Indeed, several of the major SCFA producers harboring regulatory properties, such as *F.prausnitzii*, *Prevotella*, and *Butyricimonas*, have been shown as decreased in the gut of MS. Moreover, the decrease in *Prevotella* has been associated with a Th17 expansion [45]. Other decreases in regulatory bacteria such as *Parabacteroides* or *Adlercreutzia* seem to occur. *Parabacteroides* can produce a compound called lipid 654, a TLR2 ligand significantly reduced in serum samples from MS patients compared with healthy subjects. This compound could be involved in the activation and regulation of immune responses, maintaining a certain level of TLR-2 and IFN-β signaling [77]. In addition, *Adlercreutzia* can process dietary phytoestrogens into monomeric compounds, decreasing oxidative stress and inflammatory cytokines, such as chemo-attracting proteins-1 (major monocyte recruiters) and IL-6, presenting high levels in MS [78]. Finally, the decrease in Bacteroides could decrease the induction of Treg.

In Miyake et al.’s study (Table 1), 14 species from the *Clostridium* genus (XIVa and IV clusters) were reduced in the MS patient populations [5]. Hence, such decline could lead to a deficiency of SCFAs, which are mostly produced by *Clostridium* species, and hence influence Treg promotion in proximal regions or IL-10 release [79].

On the other hand, increased genera such as *Methanobrevibacter* or *Akkermansia* and several Proteobacteria appear to be inflammation promoters. While *Methanobrevibacter* activates dendritic cells [80] and is associated with shorter time to relapse in a pediatric study [46]; *Akkermansia* is involved in the degradation process of the mucus layer, resulting in increased exposure of the resident immune cells to microbial antigens [81]. The increase in *Akkermansia municiphila* and *Acinetobacter calcoaceticus* found in a group of MS patients provoked proinflammatory responses in human peripheral blood mononuclear cells. Particularly, the in vitro results suggested that the MS-associated *Akkermansia municiphila* enhances the growth of Th1 cells from human T lymphocytes [50]. Finally, the increase in segmented filamentous bacteria and Enterobacteriaceae can orient to a Th17 immune response.

Altogether, these studies confirm a dysbiotic state in the gut of MS patients with a decrease in regulatory bacteria favoring proinflammatory pathobionts. This state is involved in a low-grade inflammation process that is constantly present during the course of this inflammatory disease. Finally, although the scientific community grew more and more results regarding gut microbiome alteration in MS, the last tend to be inconsistent. These differences are likely due to the naturally high interindividual variability, as well as to the fact that there is no real consensus in the analysis techniques (16s rRNA hypervariable regions, references, databases). In addition, microbiome analysis suffers from many confounding factors. Indeed, the content of the microbiome naturally evolves with many variables such as age, dietary habits, and medications. In order to take into consideration, all these sources of variability, very stringent study designs embedding larger sample sizes should be implemented. To reduce confounding factors, two studies emphasized the interest of using household paired design studies. Indeed, the International Multiple Sclerosis Microbiome Study consortium (iMSMS) highlighted that household is the first source of variability in the microbiome composition [69], while another study [51] included twins discordant for the disease, enabling them to account for both the environmental and genetic factors.

## 3. IgA Interactions with Microbiota and IgA-Coated Microbiota in Multiple Sclerosis

### 3.1. IgA Reciprocal Interactions with Microbiota

The main immunoglobulin class involved in responses to microbiota is IgA. Indeed, IgA is the most produced immunoglobulin of the human body, playing an important role in immune defense at mucosal surfaces (gut, lung, oral cavity, genitourinary tract, eyes, milk). The interactions between secreted IgA and the microbiota are important and reciprocal [82,83]. While in germ-free mice, IgA production is very low [83], the presence of microbiota induces the production of mucosal IgA [84] (Figure 2).

The lack of IgA results in perturbations of gut microbiota in IgA deficiency patients [85,86,87,88,89] and gut inflammation [90]. In IgA knock-out mice, the inflammation of the ileum is associated with skewed gut microbiota and is dependent on the variable region of IgA [91]. AID−/− mice with IgA deficiency exhibit a dramatic increase in segmented filamentous bacteria, whereas the reconstitution of IgA production in AID−/− mice induced recovery of normal flora [92]. IgA secretion limits the expansion of several bacteria and regulates microbial composition. IgA, which is differentially distributed between the systemic and mucosal immune systems, plays a key role in immune protection [93]. While high-affinity IgA antibodies (from T cell-dependent pathways) are thought to protect intestinal mucosal surfaces against invasion by pathogenic microorganisms, low-affinity IgA antibodies (from T cell-independent pathways) are important to confine commensal bacteria to the intestinal lumen [94]. IgA recognizes various antigens and has the ability to bind to diverse commensal bacteria [95]. IgA is also classically known for neutralizing toxins and pathogens at mucosal surfaces [96,97]. Indeed, they are able to neutralize SARS-Cov-2 [98,99] and limit its invasion [100,101].

Interactions between IgA and microbiota also occur via bacterial metabolites impacting IgA production [102]. The metabolite acetate (one of SCFAs) induces an increase in IgA production and modifications of IgA reactivity to bacteria [103]. This effect is indirect, modifying the interaction between epithelial cells and immune cells. Moreover, IgA binding to bacteria modifies their function and metabolism [104]. This illustrates how complex, reciprocal, and parallel interactions between IgA and microbiota are.

### 3.2. IgA-Coated Bacteria in Multiple Sclerosis

Linked to the altered gut microbiota composition in MS, the total amount of gut IgA-coated bacteria is decreased in MS patients compared to healthy individuals [73,105]. The more expanded disability status score (EDSS) increases, the more amount of gut IgA-coated bacteria decreases [73]. It appears that IgA production is altered in the gut of MS patients, or IgA reactivity is modified. Interestingly, in the experimental autoimmune encephalomyelitis (EAE) model, the mice exhibit a lower number of IgA+ plasma cells in their gut [105]. On the other hand, gut IgA-coated bacteria composition is also modified in MS patients [73,106]. The abundance of *Eggerthella lenta*, *Ruminococcus*, *Faecalibacterium prausnitzii*, *Pseudomonas*, *Rothiamucilaginosa*, *Blautia*, *Streptococcus*, *Clostridium*, *Eubacterium*, *Akkermansia muciniphila*, and *Bifidobacterium longum* were increased in MS IgA-coated microbiota, whereas *Anaerostipes*, *Coprococcus*, *Bacteroides*, *Lachnospira*, *Holdemania*, *Dorea*, and *Bifidobacteirum adolescentis* were decreased. Interestingly, in serum from MS patients, IgG response to commensal bacteria partially compensates for the lack of IgA responses [73]. Moreover, IgG responses against *Akkermansia muciniphila*, one of the bacteria with an increased abundance in MS gut microbiota (Table 1), are increased in blood [73] and in the cerebrospinal fluid of MS patients [107,108] and correlate with EDSS [107]. This suggests that the equilibrium between gut microbiota and IgA production is impaired in MS, inducing potential pathogenic IgG responses against gut bacteria. Moreover, it could also promote T cell responses against bacteria as autoreactive T cells in MS patients, targeting HLA-DR15 (DR2a and DR2b) molecules, cross-react with a peptide of GDP-I-fucose synthase from *Akkermansia muciniphila* [109]. In a recent study, a microRNA miR-30d was found to be increased in feces from MS patients and EAE mice [110]. This miR-30d regulates lactase from *Akkermansia muciniphila* resulting in its abundance increase and expansion of Treg [110], showing potent anti-inflammatory properties of *Akkermansia muciniphila*. Complex regulations occur in gut microbiota and antibodies responses, and further studies are needed to decipher these mechanisms in MS.

In the CNS, IgA concentrations in cerebrospinal fluid (CSF) from MS patients are increased as IgG and IgM, probably due to the breakdown of the blood-brain barrier. Oligoclonal bands of IgG in CSF are a well-validated diagnosis marker of MS and activity disease [111], but oligoclonal bands of IgA can also be found in MS CSF [112]. In addition, IgG and IgM, IgA can be produced intrathecally by plasma cells in MS patients [106,113,114,115]. However, IgA+ plasma cell frequency is lower compared to IgG+ plasma cells in MS CSF, and clonal expansion of IgA+ B cells is non-frequent [116,117]. The presence of IgA in CSF seems to be associated with less severity of disease [113,118]. IgA in CSF preponderantly recognized antigens from commensal bacteria (Gram- and Gram+ bacteria) and rarely CNS antigens [106,119]. Indeed, IgA is rarely autoreactive [120]. In Theiler’s murine encephalomyelitis virus (TMEV) infection, IgA was highly present in CNS, correlated with circulating anti-TMEV IgA titers [121]. So, IgA in CNS can be specifically directed against pathogens, but, in MS, IgA present in CNS seems to recognize gut microbiota antigens.

### 3.3. Gut IgA+ IL-10+ B Cells Exhibit Regulatory Properties, Migrate to CNS and Decrease Neuroinflammation

#### IgA+ Regulatory B Cells

Regulatory plasma cells were described in vitro and in vivo, and their function is dependent on IL-10 secretion [105,122,123,124]. The expression of B-lymphocyte-induced maturation protein 1 (BLIMP1) and interferon regulatory factor 4 (IRF4) transcription factors, mainly by plasma cells, is required for this IL-10 secretion [125]. According to several studies, plasma cells constitute the main source of IL-10 in B cell compartment [122,123,124]. These studies showed that these cells secrete mainly IgM [122,123,124]. However, other groups showed that IgA+ B cells also exhibit regulatory properties [105,126,127,128]. These B cells secrete IL-10 and express PD-L1 and appear to be dependent on the proliferation-inducing ligan (APRIL) and TGFβ signaling for class switch and maturation [126,127,129]. Moreover, Tregs as dendritic cells stimulated with microbiota components can promote these regulatory IgA+ B cells [129,130]. In return, IgA+ B cells stimulated with microbiota components are able to induce functional Tregs in vitro [130]. They also inhibit T cell and macrophages responses [126].

### 3.4. Regulatory IgA+ B Cells in Multiple Sclerosis

In physiological conditions, plasma cells are absent in CNS, except in particular regions such as meninges [131,132] and choroid plexus [133]. IgA+ plasma cells, which are educated in the gut, appear to be important for immune defense in this area [134]. In addition, regulatory IgA+ B cells also appear to downregulate inflammation in CNS. This was demonstrated in mouse models. Indeed, EAE disease was ameliorated in transgenic mice for APRIL or B cell-activating factor (BAFF) (two cytokines important for the induction of IgA class switch in the gut) [105,126]. Those mice exhibit increased production of IgA and frequency of IgA+ B cells. Inversely, the knock-out of IgA heavy chain (Cα) resulted in the exacerbation of the disease [105]. The mechanisms of regulation require the secretion of IL-10 as their beneficial effects were abolished in IL-10 KO mice [105]. Moreover, Rojas et al. demonstrated that IgA+ cells, specific for commensal bacteria, can migrate to CNS, showing that IgA+ B cells can locally regulate neuroinflammation [105]. The invalidation of the BAFF and APRIL receptor called B cell maturation antigen (BCMA) in mice also resulted in the exacerbation of EAE [135].

In MS patients, IL-10+ plasma cells can be found in CNS lesions [136]. Moreover, IgA+ B cells were found in the brain of MS patients [106,137], and IgA+ IL-10+ B cells in MS lesions were recently detected [106]. Altogether, these results in humans suggest that, as in mice, bacteria-specific IgA+ IL-10+ B cells educated at mucosal surfaces can migrate into the CNS to regulate inflammation via IL-10 secretion.

Therapies targeting B cells are efficient in MS using anti-CD20 or anti-CD19 antibodies depleting B cells. These therapies probably avoid antigen B cell presentation to T cells and secretion of proinflammatory cytokines. However, the transmembrane activator and calcium modulator and cyclophilin ligand interactor-immunoglobulin (TACI-Ig or atacicept) treatment blocking APRIL and BAFF signaling and plasmablast/plasma cells maturation (Figure 3) was a failure in MS, resulting in exacerbation of relapse rate [138]. This trial confirmed that plasma cells in MS are essential for regulation, and their lack is deleterious for MS patients.

## 4. Treatments Targeting IgA and Microbiota

As microbiota and IgA are altered in MS, therapies that target this interactive system are promising. Therapeutic medicines targeting the gut microbiota can have a big impact on disease progression and symptom management [139]. To influence the microbiome and modulate immune responses, antibiotics, colonization with single or multiple microbial species, fecal transplantation, and therapy with microbial antigens and metabolites are used. There is currently no FDA-approved MS medication that targets the gut microbiome. However, ongoing trials and case reports bring new data supporting new therapeutic avenues. In this part of the review, we will focus the discussion on animal and human tests targeting microbiota and IgA.

### 4.1. Modifications of Gut Microbiota in EAE

Interventions to modify gut microbiota were deeply studied in EAE. Indeed, the drastic depletion of gut microbiota resulted in amelioration of neuroinflammation. The germ-free (GF) mice have a lower proportion of pathogenic Th17 cells in the intestinal lamina propria, which helps to downregulate spontaneous EAE [140]. Furthermore, B cell responses have also been shown to be suppressed [140]. GF mice produce more CD4+CD25+FoxP3+ Tregs, and GF immunized mice generate relatively low levels of IFNγ and IL-17A. Moreover, antibiotics can deeply alter the microbiome’s composition. Oral antibiotics have been proven in numerous investigations to slow the progression of EAE by disrupting gastrointestinal symbiosis [141]. Using an oral wide spectrum antibiotic lowers the severity of EAE by boosting Treg cells and IL-10 while decreasing IFNγ, TNFα, IL-6, and IL-17 [142]. The development and severity of disease are influenced by the unique relationships of GF mice with identified commensal bacterial species. As previously stated, the colonization of GF mice with segmented filamentous bacteria leads to an increase in Th17 cell population and IL-17 accumulation in the gut, which favors Th17 proliferation in the spinal cord and the development of EAE [143]. *Bacteroides fragilis*, for example, produces polysaccharides, which increase the number of intestinal Tregs and prevent mice from developing CNS autoimmunity by inducing tolerogenic CD103+ dendritic cells in CNS local lymph nodes [144]. Fecal microbiota transplantation (FMT) from MS patients into GF mice induced exacerbation of EAE, Th17 responses, and decrease in IL-10 production [50,51]. By contrast, FMT from healthy donors resulted in regulatory immune responses.

The MS pathology has been linked to viral illnesses such as Epstein Barr virus (EBV) or human herpes virus 6 (HHV6) infections. Immune responses to viral infection may be influenced by the gut microbiome. It has been discovered that intracranial infection with Theiler’s virus (TMEV) affects gut microbiota in the MS Theiler’s viral model, which is a unique example of how a pathogen implanted in the brain produces inflammation, demyelination, autoimmune, and neuronal damage. Depletion of microbiota by oral antibiotic treatment, on the other hand, modifies neuroimmune responses to TMEV during the acute phase of the disease, providing another example of the bidirectional communication system occurring along the gut-brain axis [145,146].

Interestingly, supplementation of probiotics to the EAE mice model enhances their therapeutic grade by increasing the production of IL-10-producing Tregs [147]. Recent research has confirmed the importance of CD39+Foxp3+ regulatory T cells in IL-10-positive effects. The integrity and function of brain biological barriers are regulated by gut microbiota, which is important because of the breakdown of the MS blood-brain barrier [148,149].

### 4.2. Effects of Disease-Modifying Therapies on MS Gut Microbiota

In humans, the changes in the microbiota of MS patients have been linked to disease status, disease-modifying therapies, and immunologic changes in multiple investigations. In a group of patients, there were significant increases in *Akkermansia muciniphila* and *Methanobrevibacter smithii*, as well as a decrease in *Butyricimonas*. More importantly, the above changes were reversed in patients receiving disease-modifying treatments [44], suggesting that MS therapies targeting the immune system may correct microbiota alterations (Table 1). Indeed, dimethyl fumarate (DMF) [61,64] and interferon β (IFNβ) [76] can be associated with a normalization of microbiota genera and an increase in *Faecalibacterium* in MS patients. In the study of Chen et al. (Table 1), the gut microbial community of treated RRMS patients was more similar to healthy controls than the gut microbiota of RRMS patients with active disease, according to gut community research. *Adlercrutzia*, *Collinsella*, *Parabacteroides*, *Coprobacillus*, *Lactobacillus*, and *Haemophilus* were shown to have considerably lower abundances in MS patients at the genus level. In MS patients, however, the bacteria *Flavobacterium*, *Pedobacter*, *Blautia*, *Dorea*, *Mycoplana*, *and Pseudomonas* were shown to be considerably higher [45]. However, other disease-modifying therapies were shown to have different effects. Indeed, glatiramer acetate (GA) and fingolimod (FG) increased atypical forms of *E. coli*, *Enterobacter*, fungi of the genus *Candida*, *Proteus*, and *Parvimonas micra*, also suggesting a negative impact of these treatments on the intestinal flora [55]. As a result, modifications of immune responses by disease-modifying therapies induce various modifications of gut microbiota.

### 4.3. Treatments Targeting Gut Microbiota in Multiple Sclerosis Patients

It is highly possible that diet can also modulate the observed dysbiosis in MS. Indeed, it was shown that a high vegetable diet can increase *Lachnospiraceae* and *Roseburia* levels in MS and correlates with several immune regulatory components, such as a positive correlation with IL-10 and Tregs and a negative correlation with the production of proinflammatory interleukin IL-17, compared to western diet [52]. Moreover, a ketogenic diet, consisting of a high-fat/low-sugar diet, increased microbiome diversity in MS, often associated with health [48]. Similar results were obtained applying intermittent fasting [54].

Fecal microbiota transplant (FMT) deeply modulates gut microbiota ecology. One case report claims beneficial effects of FMT in MS patients treated for recurrent *Clostridioides difficile* infection [150]. They observed the resolution of *Clostridioides difficile* infections and the amelioration of MS symptoms. A pilot study tested FMT in one MS patient and examined its gut microbiota [68]. FMT was associated with the normalization in the altered microbiota genera and an increase in *Faecalibacterium* abundance. There are three ongoing trials with FMT (Table 2).

### 4.4. Neurofilament Light Chain, PBMC: Peripheral Blood Mononuclear Cells

To avoid the drastic changes in gut microbiota with FMT and risk of infection [151], probiotic therapies started to be tested in MS. In a first study, 20 RRMS patients received probiotics (three strains of *Lactobacillus* and one of *Bifidobacterium*), and 20 received placebo [152]. The probiotic supplementation resulted in downregulation of proinflammatory cytokines. In additional randomized, double-blinded studies, inflammatory circulating factors and the EDSS were reduced in treated MS patients [153] as the mental behavior [154]. Another pilot study investigated the effects of VSL3, a cocktail of eight bacteria (four strains of *Lactobacillus*, three strains of *Bifidobacterium*, and one strain of *Streptococcus*) in RRMS patients showed modifications of gut flora and circulating immune cells [53]. After treatment, an increase in *Lactobacillus* and a decrease in *Akkermansia* and *Blautia* were observed. Not only bacteria are considered probiotic but also helminths and fungi. Two trials were performed using helminth [155,156,157] and one using *Saccharomyces boulardii* [158]. Although these preliminary trials suggest new approaches to the therapy of MS (Table 2), our current review is mostly restricted to ongoing trials, whose benefits remain to be proven.

### 4.5. Therapeutic Interventions on IgA+ B Cells

Disease-modifying therapies, as they directly target immune cells, could impact IgA production. Indeed, IgA+ plasma cells in mice are sensitive to sphingosine 1-phosphate receptor modulation [159,160,161]. Fingolimod treatment, which targets this receptor, resulted in the reduction in B cells and IgA production in mice. Secretory IgA concentrations in stools and saliva were equivalent in MS patients treated with fingolimod compared to GA-treated MS patients [162]. A comparison with untreated patients would be very informative. So, these disease-modifying therapies could negatively impact regulatory IgA+ B cells in MS patients. A way to restore these Breg cells could be to use cytokines that promote their stimulation and proliferation, such as BAFF. In BAFF transgenic mice, an increase in IgA+ Breg is observed [105]. Susceptibility in EAE is associated with low circulating BAFF level [163]. However, intracerebral BAFF levels are associated with the formation of lymphoid structure in mice with active disease [164] and activation of B cells [165]. In MS patients, higher BAFF expression in CSF was associated with cortical damages [166]. Moreover, BAFF induced a proinflammatory phenotype on microglia [167]. The stimulation with BAFF on B cells from MS patients in vitro does not induce IL-10 producing B cells [168]. BAFF is overexpressed in MS patients [169] due to the polymorphism of its gene (TNFSF13B). As BAFF signaling is important for various cells, BAFF treatment to induce Breg appears to be compromised. Moreover, disease-modifying therapies seem to decrease BAFF levels in MS [170]. More targeted therapies on these Breg are needed.

Another way to indirectly stimulate these Breg is to target microbiota. Food, nutrient intake, and prebiotics are known to modify gut microbiota, but they also impact mucosal IgA secretions. High-fat diet results in the alteration of intestinal IgA+ B cells and IgA production [171]. Metformin treatment partially restores IgA+ B cells in mice, and SIgA concentrations increase after bariatric surgery in obese patients [171]. Dietary restriction promotes IgA production [172] as dietary fiber [173,174] or polyphenol supplementation [175]. Dietary supplementation with bacterial components from *Lactobacillus* such as KDP [176] or EPS [177] also increases gut IgA production. Concerning probiotics, their oral administration (*Enterococcus faecium* [178], *Bifidobacteirum bifidum* [179], *Enterococcus durans* [180], *Lactobacillus paracasei* [181]) also results in augmentation of intestinal IgA secretions. Probiotics can be combined with resveratrol showing synergic effects [182]. They can be embedded in the yeast membrane to be directly delivered to Peyer’s patches inducing immune responses and secretion of IgA [183]. Lastly, metabolites derived from bacteria production are involved in this process. Bacterial butyrate promotes gut T cell-independent IgA responses [184]. Acetate promotes IgA class switch via dendritic cells [185].

## 5. Conclusions

IgA and microbiota are altered in MS patients by complex mechanisms. IL-10+ IgA+ regulatory B cells, specific from gut commensal bacteria, are important immune players, potentially regulating neuroinflammation. Targeting microbiota with probiotics or metabolites to increase these regulatory B cells seems to be promising in MS.

## Figures and Tables

**Figure 1 microorganisms-10-00617-f001:**
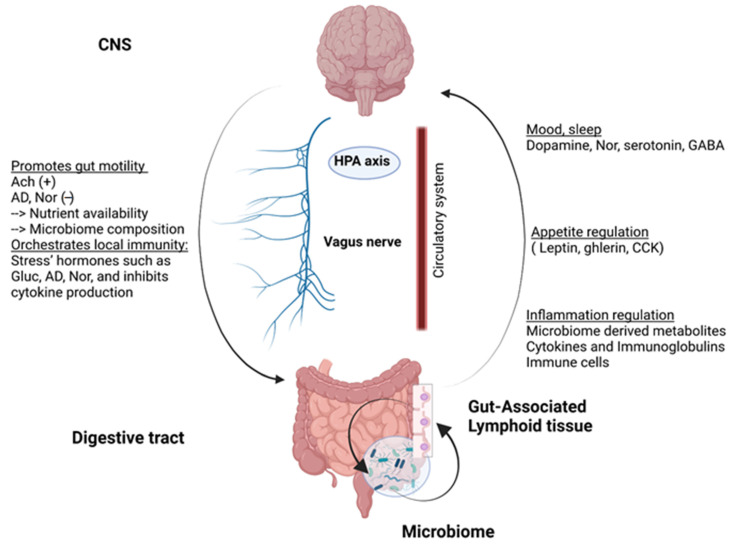
Gut-brain axis, a multidirectional communication system: Communications take place through the neuroendocrine pathway. While acetylcholine (Ach) promotes smooth muscle contractions, adrenergic neurons can decrease bowel movement. Moreover, stress hormones, mainly glucorticoĩds (Gluc), adrenaline (AD), and noradrenaline (Nor), create a strong suppressive response on the immune system. On the other side, the digestive system can release large amounts of bioactive hormones and molecules in cooperation with the constantly interacting microbiome and immune system. HPA: hypothalamic-pituitary-adrenal axis, GABA: γ-aminobutyric acid, CCK: cholecystokinin.

**Figure 2 microorganisms-10-00617-f002:**
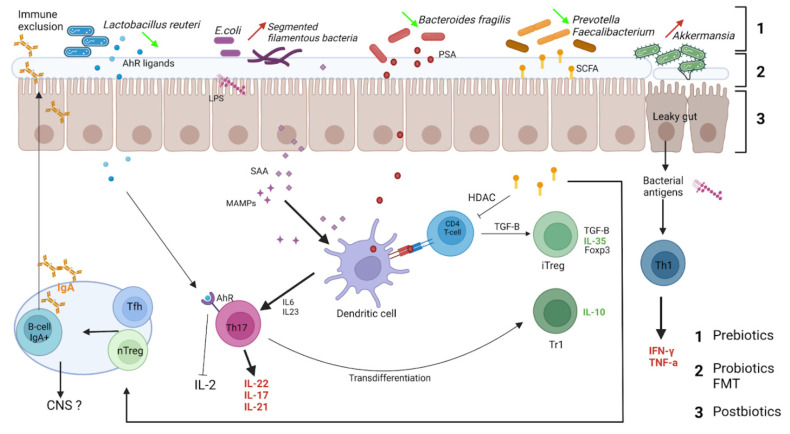
Relevant interactions between microbiota and intestinal immune cells in the context of multiple sclerosis. Gut dysbiosis in MS may participate in the inflammatory environment. Bacteria providing inflammatory signals are increased at the expense of bacteria species harboring regulatory properties. Immune cells arising from the gut can then access extraintestinal tissues and regulate CNS inflammation. Tfh: follicular helper T cells, nTreg: natural regulatory T cell, AhR: aryl hydrocarbon receptor, LPS: lipopolysaccharide, SAA: serum amyloid A, MAMPs: microbiome-associated molecular patterns, PSA: polysaccharide A, SCFA: short-chain fatty acids. Dietary interventions could mitigate these inflammatory signals: 1: Prebiotics, consisting of insoluble fibers, promote the growth of regulatory bacteria. 2: Probiotics can directly normalize local flora. 3: Postbiotics: bioactive molecules secreted by the microbiota activity such as SCFA, glutamine promotes integrity of the epithelial barrier.

**Figure 3 microorganisms-10-00617-f003:**
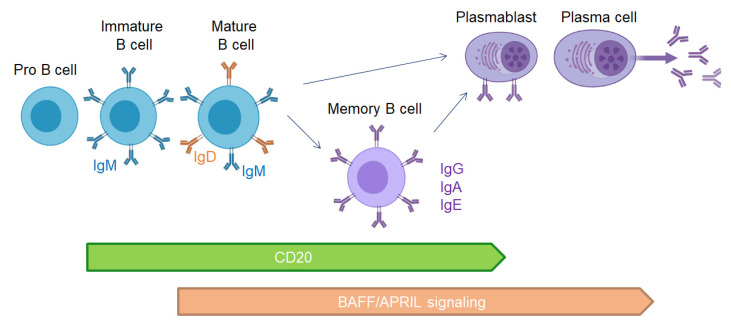
CD20 expression and BAFF/APRIL signaling during B cell development.

**Table 1 microorganisms-10-00617-t001:** Studies investigating the gut microbiome alterations in multiple sclerosis patients.

Study	Country	Study Design	Profiling Method	Main Results
Cantarel et al., 2015 [6]	USA	5 RRMS/8 HV Stools	16S rRNA-DNA chip hybridization	β diversity: NA, α diversity: NA Up: *Ruminococcus* Down: *Faecalibacterium*, *Bacteroides*
Miyake et al., 2015 [5]	Japan	20 RRMS/40 HV Stools	16S rRNA V1–V2 (pyrosequencing)	β diversity: *p* < 0.05 α diversity: NS up: *Streptococcus*, *Eggerthella* down: *Faecalibacterium*, *Prevotella*, *Anaerostipes*
Jangi et al., 2016 [44]	USA	60 RRMS/43 HV Stools	16S rRNA V3–V5	β diversity: NS, α diversity: NS Up: *Akkermansia*, *Methanobrevibacter* Down: *Butyricimonas*, *Collinsella*, *Slackia*, *Prevotella*
Chen et al., 2016 [45]	USA	31 RRMS/36 HV Stools	16S rRNA V3–V5	β diversity: *p* < 0.001, α diversity: NS Up: *Pseudomonas*, *Mycoplana*, *Haemophilus*, *Blautia*, and *Dorea* Down: *Parabacteroides*, *Adlercreutzia* and *Prevotella*
Tremlett et al., 2016 [46]	USA	18 RRMS/17 HV (pediadric) Stools	16S rRNA V4	β diversity: NS, α diversity: NS Up: Desulfovibrionaceae (Bilophila, Desulfovibrio and Christensenellaceae) Down: *Lachnospiraceae* and *Ruminococcaceae*
Cree et al., 2016 [47]	USA	16 RRMS/16 HV Stools	DNA hybridization PhyloChip G3–entire 16S rRNA	NS
Swidsinski et al., 2017 [48]	Germany	25 RRMS/14 HV Stools	FISH to specific 16S rRNA probes	β diversity: *p* < 0.001, α diversity: NS, decreased bacterial mass Down: *Roseburia*, *Bacteroides*, *Faecalibacterium prausnitzii*
Cosorich et al., 2017 [49]	Italy	19 RRMS/17 HV Small intestine biopsies	16S rRNA V3–V5	β diversity: NA, α diversity: NS Up: *Firmicutes/bacteroides* ratio, *Streptococcus*, Down: *Prevotella*
Cekanaviciute et al., 2017 [50]	USA	71 RRMS/71 HV Stools	16S rRNA V4	β diversity: NS, α diversity: NS Up: *Akkermansia*, *Acinetobacter*, *calcoaceticus* Down: *Parabacteroides*
Berer et al., 2017 [51]	Germany	34 twin pairs (MS/HV) 3 CIS, 22 RRMS, 7 SPMS, 2 PPMS Stools	16S rRNA V3–V5 + shotgun sequencing	β diversity: NS, α diversity: NS Up: *Akkermansia*
resella et al., 2017 [52]	Italy	20 RRMS, 2 diets (10 Western/10 high-fiber diet) Stools	16S rRNA V4	α, β diversity: NS Up *Lachnospiraceae Coprococcus eutactus*, *Ruminococcus lactaris*, *Roseburia intestinalis*, *Hungatella* Down: Th17, correlation *Lachnospiraceae*, and Treg
Tankou et al., 2018 [53]	USA	9 RRMS/13 HV Stools	16S rRNA V4	Probiotics decreased *Akkermansia* and *Blautia*, increased *Lactobacillus* in MS
Cignarella et al., 2018 [54]	USA	8 RRMS before/after intermittent fasting Stools	16S rRNA V1–V3	Adiponectin levels correlate with *Faecalibacterium*
Abdurasulova et al., 2018 [55]	Russia	17 RRMS GA/17 RRMS FG Stools	16S rRNA	α, β diversity: NA Up: atypical *E coli*, *Enterobacter* sp. with GA treatment Down: normal *E coli*
Forbes et al., 2018 [56]	Canada	19RRMS, 21RA, 20CD, 19UC, 23HV	16S rRNA V4	β diversity: NA α diversity: NS Up: *Actinomyces*, *Eggerthella*, *Anaerofustis*, *Clostridia XIII*, *Clostridium III*, *Faecalicoccus*, *Streptococcus* Down: *Butyricicoccus*, *Faecalibacterium*, Dialister, *Gemmiger*, *Lachnospiraceae*, Subdolibacterium MS signature partially overlays other inflammatory disease signatures
Nourbakhsh et al., 2018 [57]	USA	Pediatric cohort	16S rRNA V4	Same cohort than Tremlett et al., 2016
Tankou et al., 2018 [58]	USA	RRMS/HV	16S rRNA V4	Same cohort than Tankou et al., Ann Neurol 2018
Tankou et al., 2018 [53]	USA	9 RRMS (2+GA) 13HV Before and after VSL3 probiotics Stools	16S rRNA V4	Up with VSL3: *Lactobacillus* Down with VSL3: *Akkermansia*, *Blautia* Decrease in inflammatory monocytes markers of cell activation
Zeng et al., 2019 [59]	China	34 RRMS/34 NMO/12HV Stools	16S rRNA V3–V4 + metabolomic	β diversity: *p* < 0.01, α diversity: NS Up: *Streptococcus* Down: *Prevotella*, *Faecalibacterium* + Decreased SCFA in MS and NMO
Oezguen et al., 2019 [60]	USA	13 RRMS/14 HV Stools	16S rRNA V3–V5	β diversity: NA, α diversity: up *p* = 0.04 Down: *Prevotella*
Storm-Larsen et al., 2019 [61]	Norway	27 RRMS +DMF, 9 RRMS +GA or IFN-𝛽 before and after treatment 165 HV Stools	16S rRNA V3–V4	β Diversity: *p* = 0.01, α diversity: NS At baseline in MS vs. HV: Down: *Faecalibacterium* Up: *Faecalibacterium* with DMF treatment
Kozhieva et al., 2019 [62]	Russia	15 PPMS/15 HV Stools	16S rRNA V3–V4	β diversity: NA, α diversity: increased richness, *p* < 0.05 Up: *Gemmiger*, *Ruminococcus*
Ventura et al., 2019 [63]	USA	40 RRMS/41 HV Stools Multiethnic ancestry	16S rRNA V4 + shotgun sequencing (24 RRMS/24 HV)	β diversity: *p* < 0.01 in Hispanic ancestry α diversity: NS Up: *Clostridium* in all ancestries, *Akkermansia* in European ancestry Down: *Prevotella* in Hispanic ancestry
Katz Sand et al., 2019 [64]	USA	75 untreated RRMS 33 + DMF 60 + GA Stools	16S rRNA V4	β diversity: NS, α diversity: NS Up: *Lachnospiraceae and Veillonellaceae* with all treatments Down: *Firmicutes*, *Fusobacteria*, *Clostridiales* with DMF
Choileáin et al., 2020 [65]	USA	26 RRMS/39 HV Stools	16s rRNA V4	β diversity: *p* < 0.01, α diversity: down (Shannon, *p* < 0.05) Up: *Bacteroidetes* Down: *Coprococcus*, *Firmicutes*, *Paraprevotella*, *Ruminococcaceae*
Saresella et al., 2020 [66]	Italy	26 RRMS/12 SPMS/38 HV Stools	16S rRNA V3–V4 + metabolomic	β diversity: *p* < 0.0001, α diversity: NS Up: *Akkermansia* in SPMS, *Streptococcus* in RRMS, *Collinsella* in RR and SPMS. Serum caproic acid Down: *Coprococcus*, *Roseburia* in RR and SPMS, *Lachnospira* in RRMS. Serum butyric acid
Takewaki et al., 2020 [67]	Japan	62 RRMS/15 SPMS/22 “atypical” MS/20 NMO/55HV Stools	16S rRNA V1–V2	β diversity: RRMS, SPMS and NMO vs. HV: *p* < 0.05 α diversity: NS Up: *Akkermansia in RRMS* vs. *HV. Streptococcus*, *Clostridium* in RRMS and SPMS vs. HV Down: *Eubacterium*, *Lachnospiraceae*, *Megamonas* in RRMS vs. HV
Engen et al., 2020 [68]	USA	1 RRMS before and after FMT Stools	Shotgun sequencing	β diversity: NA α diversity: FMT increased alpha diversity indices. Up: FMT increased *Faecalibacterium Prausnitzii*
The iMSMS Consortium, 2020 [69]	International	128 MS/128 HV Household paired Stools	16S rRNA and Shotgun sequencing	Emphasize the importance of paired household design to reduce interindividual variability
Ling et al., 2020 [70]	China	22 RRMS/33 HV Stools	16S rRNA V3–V4	β diversity: NS, α diversity: NS Up: *Blautia*, *Flavonifractor* Down: *Faecalibacterium*, *Granulicatella*, *Prevotella*, *Roseburia*
Reynders et al., 2020 [71]	Belgium	24 untreated RRMS/26 PPMS/20 benign MS, 24, INF𝛽 treated/4 RRMS relapse/120HV Stools	16S rRNA V3–V4	β diversity: NA, α diversity: Down in IFN β treated and relapse RRMS compared to benign forms Up: *Alistipes*, *Anaerotruncus*, *Clostridium cluster IV*, *Lactobacillus*, *Methanobrevibacter*, *Olsenella*, *Parabacteroides*, *Ruminococcus*, *Sporobacter* in MS vs. HC Down: *Butyricicoccus*, *Gemmiger*, *Intestinibacter*, *Roseburia* in MS vs. HC, *Butyricicoccus* in PPMS vs. RRMS
-Cox et al., 2021 [72]		199 RRMS/44 progressive MS/40 HV Stools	16S rRNA V4	β diversity: *p* < 0.01 RRMS vs. HV, *p* < 0.01 PPMS vs. HV α diversity: increased Shannon and richness in RR and PPMS compared to HV Up: *Clostridium*, *Bacteroides*, *Gemella*, *Akkermansia* in RR and progressive MS Down: *Prevotella* in RRMS, *Dorea* in RR and progressive MS
Sterlin et al., 2021 [73]		30 RRMS/15 CIS/32 HV Stools	16S rRNA V3–V5	β diversity: NA, α diversity: NS Commensal-specific gut IgA responses are drastically reduced in MS patients with severe disease
Jenkins et al., 2021 [74]		50 RRMS parasite challenge, 24 treated with antiparasitic agent 26 untreated	16S rRNA V3–V4	Up: *Parabacteroides* in treated RRMS Down: *Roseburia*, *Dorea*, *Tyzzerella*, *Anaerostipes*, *and Agathobacter* in treated RRMS compared to placebo RRMS No relapse in treated group
Levi et al., 2021 [75]		129 RRMS/58 HV Stools	Shotgun sequencing	β diversity: NA, α diversity: NA Up: *Lawsonella* Down: *Faecalibacterium* *prausnitzii*, *Bacteroides fragiils*, *Eubacterium rectale*, *Butyrivibrio*, *Clostridium*, *Coprococcus*, *Roseburia*
Castillo-Álvarez et al., 2021 [76]	Europe (Spain)	15 RRMS IfNb/untreated 15RRMS/14 HV Stools	16S rRNA V4	β diversity: NS, α diversity: down (*p* = 0.08) Up: *Faecalibacterium*, *Ruminococcus*, *Blautia*, *Anaerostipes*, *Bifidobacterium* Down: *Prevotella* IFN β partially restored microbiota

FISH: fluorescence in situ hybridization, NS: non-significant, NA: not available, DMF: dimethyl fumarate, FG: fingolimod, IFNβ: interferon-β, GA: glatiramer acetate, RA: rheumatoid arthritis, CD: Crohn’s disease, UC: ulcerative colitis, FMT: fecal microbiota transplant.

**Table 2 microorganisms-10-00617-t002:** Reported clinical trials assessing the effects of gut microbiota modulation in multiple sclerosis.

Trial ID	Design	Intervention	Outcome Variables	Status
NCT03183869	Prospective interventional randomized crossover Assignment	FMT	25 cytokines levels in blood	Dropped
NCT03594487	Prospective interventional non-randomized parallel assignment	FMT	Engraftment, safety, Ig levels, B cell counts, MRI new T2 lesion incidence	Ongoing
NCT04150549	Prospective interventional randomized parallel assignment	FMT	Changes in T2 Lesions-MRI	Ongoing
NCT04574024	Prospective interventional randomized parallel assignment	High-fiber supplementation	Composition of gut microbiota, production of SCFAs and Tregs	Ongoing
NCT04038541	Prospective interventional randomized crossover assignment	Prebiotic vs. probiotic	Changes in gene expression in PBMCs, Nfl concentration, gut microbiota in stool samples	Ongoing
NCT01413243	Prospective interventional randomized parallel assignment	Trichuris suis ova probiotic	Incidence of new T2 lesion-MRI	Terminated
NCT04599595	Observational prospective	Transanal irrigation	Incidence and prevalence of intestinal dysfunction in multiple sclerosis	Terminated

FMT: fecal microbiota transfer, MRI: magnetic resonance imaging, Ig: immunoglobins, Nfl: neurofilament light chain.

## Data Availability

Not applicable.

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
