# Peer review of "Microbiota, IgA and Multiple Sclerosis"

_microorganisms, 2022, doi:10.3390/microorganisms10030617_

Round 1

Reviewer 1 Report

This is thoughtfully organized review with good insights into the current understanding of how the gut microbiome modulates CNS inflammation/MS and the role of IgA-coated gut microbiota in MS. I have only minor suggestions for improvement.

1) there are many grammatical errors typos  - please check carefully. e.g. line 169, "Moveover...?", line 115 "Disturbances...dysbiosis", line 43 -"interphase" - should be "interface".

2) some acronyms (such as BCMA, TACI, etc) need to be defined.

3) line 320 -" the invalidation..." this sentence is confusing. please clarify.

4) line 350 -  GF mice do not have " a higher proportion of Tregs" especially in the intestine. this is a misleading statement.

5) the image resolution of the figures is very poor.

Author Response

We would like to thank the reviewers for the time spent and for their pertinent feedbacks allowing us to improve our manuscript.

Reviewer 1:

  • there are many grammatical errors typos  - please check carefully. e.g. line 169, "Moveover...?", line 115 "Disturbances...dysbiosis", line 43 -"interphase" - should be "interface".

Thank you for the remark. We took into account these corrections and included them in the revised manuscript.

      2) some acronyms (such as BCMA, TACI, etc) need to be defined.

We have carefully reviewed our manuscript and added all the missing abbreviations

  • l320 -" the invalidation..." this sentence is confusing. please clarify.

We added the required precisions in the sentence: l234: “the targeted invalidation of CD-19 using conditional knockout impacted IgA B cells and resulted in the exacerbation of the disease”

  • line 350 -  GF mice do not have " a higher proportion of Tregs" especially in the intestine. this is a misleading statement.

We thank the review for detecting this error. There was a mix between GF mice and GF mice receiving microbiota from healthy donors (reference 51, Berer et al PNAS 2017). The reference was not the good one, we wanted to mention the study from Berer et al Nature 2011. We modified the references. We apologize and have corrected the sentence. “The germ free (GF) mice have a lower proportion of pathogenic Th17 cells in the intestinal lamina propria, which helps to downregulate spontaneous EAE (Ref Berer et al 2011 Nature). Furthermore, B cell responses have also been shown to be suppressed (Ref Berer et al 2011 Nature).”

  • the image resolution of the figures is very poor.

Thank you for letting us know. It seems the pdf exportation process has greatly degraded the image quality. We performed the building of PDF to recover the high-resolution figures for the final version.

Reviewer 2 Report

Dear Authors,

I have read with great interest your review article titled: Microbiota, IgA and Multiple Sclerosis. It is indeed a well-written and conducted review with sufficient information. However, the quality of the figures should be improved. Careful on the typos since I identified several ones and how abbreviations were used. For this review, I would advise being added a Methodology section. There is no searching strategy, keywords used for studies identification, databases used to identify suitable articles, inclusion/exclusion criteria, and limitations of the study.

  Kind regards,   The Reviewer

Author Response

We would like to thank the reviewers for the time spent and for their pertinent feedbacks allowing us to improve our manuscript.

  • I have read with great interest your review article titled: Microbiota, IgA and Multiple Sclerosis. It is indeed a well-written and conducted review with sufficient information. However, the quality of the figures should be improved. Careful on the typos since I identified several ones and how abbreviations were used. For this review, I would advise being added a Methodology section. There is no searching strategy, keywords used for studies identification, databases used to identify suitable articles, inclusion/exclusion criteria, and limitations of the study.

Thank you for the relevant points. The following elements have been added to the manuscript to take into account the raised points:

  • A short methodology paragraph was added in paragraph 2.3 (l.129-l.137) summarizing our searching strategy and added a key limitation of our clinical trial review (l.456):

(l.129-l.137); “For the purpose of this review, we carried out a systematic review of all the reported studies investigating the gut microbiome content in multiple sclerosis. To identify studies of interest, we used medical subject headings (MeSH) function in pubmed (https://pubmed.ncbi.nlm.nih.gov/). We interrogated the database using different term combinations with multiple sclerosis, microbiota, 16s rRNA, whole genome sequencing and filtered for original articles. For the ongoing clinical trials, we searched through clinicaltrials.gov (https://clinicaltrials.gov/) database with multiple sclerosis in the condition input and terms such as probiotic, prebiotic, fecal microbiota transplant, short-chain fatty acids “

l.456: “Although these beginning trials offer new perspectives to complete MS therapies (Table 2), our current review is mostly restricted on-going trials, which added benefices are still to be proven.”

  • As mentioned, the figure quality has been degraded during the pdf exportation process. We fixed this point in the revised version.
  • Missing abbreviations were added and typos errors were carefully corrected.